# Nile basin flow regimes under 21$^{st}$ century climate variability

Hesham Elhaddad [1,2], Mohamed Sultan [1]✉, Eugene Yan[3], Duc Tran [4], Hugo E. Torres-Uribe[3] & Hadi Karimi [1]

The Nile Basin, a freshwater resource for over 300 million people, faces unprecedented hydrological risks under climate change and transboundary water disputes. Yet, basin-wide projections of extreme streamflow events remain limited by fragmented modeling and insufficient integration of climate uncertainty. Here, we assess future flood risk in downstream countries using a calibrated, climate-driven Soil and Water Assessment Tool model, forced by bias-corrected CMIP6 models under the SSP2-4.5 and SSP5-8.5 scenarios, marking the first application of its kind targeting the Nile Basin downstream regions. Our results indicate a 63% (SSP2-4.5) and 85% (SSP5-8.5) increase in 100-year peak discharges in the 21st century, with extreme floods occurring nearly every decade under high-emission scenarios. Our climate-driven hydrologic modeling, risk analysis, and climate projections emphasize the need for coordinated planning, provide actionable risk information, and a framework for regional cooperation and preparedness to mitigate future flood risks and address water security challenges in the Nile Basin.

## Main

Ensuring water security in transboundary river basins under intensifying climate change is a significant challenge in Earth and environmental sciences, requiring holistic insights into basin-wide systems. The Nile Basin, often cited as the world's longest river system, spanning nearly three million square kilometers in eleven countries (Fig. 1a)[1], exemplifies how limited water resources can trigger geopolitical tensions. Egypt's population, which rose from ~44 million in 1980 to over 109 million by 2020[2], relies on the Nile for more than 90% of its freshwater[3]. By 2017, Egypt's annual per capita share had fallen to ~628 m³, about half of the international standard[4] of 1000 m³. Similarly, the Nile provides 73% of the annual freshwater in northern Sudan, with a nominal allocation of 18.5 km³/yr under the 1959 Nile Water Agreement[5]. Yet high evaporation (roughly 13% of the flow) and escalating demands pose ongoing challenges[5]. These pressures underscore concerns that anthropogenic forces (e.g., dams) and natural factors (e.g., climate change) could further strain downstream water availability in the 21st century, highlighting a critical gap in basin-wide projections of hydrological risks under future climate scenarios. A brief description of the geography and climate of the Nile Basin and its sub-basins is provided in Supplementary Text 1.

Several recent studies have examined the immediate effects of large infrastructures, with particular attention to the Grand Ethiopian Renaissance Dam (GERD) on the Blue Nile (BN), which supplies around 60% of the downstream Nile discharge[6]. Some predict a minimal long-term impact on water availability once the reservoir is filled[7], whereas others find significant shortfalls under different hydrological scenarios[8]. Field-based observations and satellite data (e.g., radar altimetry, the Gravity Recovery and Climate Experiment [GRACE], and GRACE Follow-On [FO]) reveal substantial seepage losses during the GERD's initial filling years, attributed to fault networks beneath the reservoir that act as preferred pathways for groundwater flow from the reservoir to the surroundings[9]. These studies are crucial for understanding the impact of infrastructure on current water management, flood control, and ecosystem restoration. On the other hand, long-term shifts in climate are typically felt on larger scales and can profoundly reshape hydrological regimes over decades[10]. Thus, for future planning, long-term sustainability, or adaptation, these shifts are typically considered the primary concern[10].

Climate models remain essential tools for assessing how future climate variability and long-term change could affect water resources in the Nile Basin. Still, projections continue to face significant uncertainties, especially for precipitation, as different models and downscaling methods often yield divergent outcomes[11]. Recent studies provide important insights but also illustrate these uncertainties. In the Blue Nile, one analysis applied a calibrated rainfall–runoff model driven by three General Circulation Models

[1]Department of Geological and Environmental Sciences, Western Michigan University, Kalamazoo, MI, USA. [2]Geodynamics Department, National Research Institute of Astronomy and Geophysics (NRIAG), Helwan, Cairo, Egypt. [3]Environmental Science Division, Argonne National Laboratory, Argonne, IL, USA. [4]Department of Civil and Environmental Engineering, The University of Virginia, Charlottesville, VA, USA. ✉e-mail: mohamed.sultan@wmich.edu

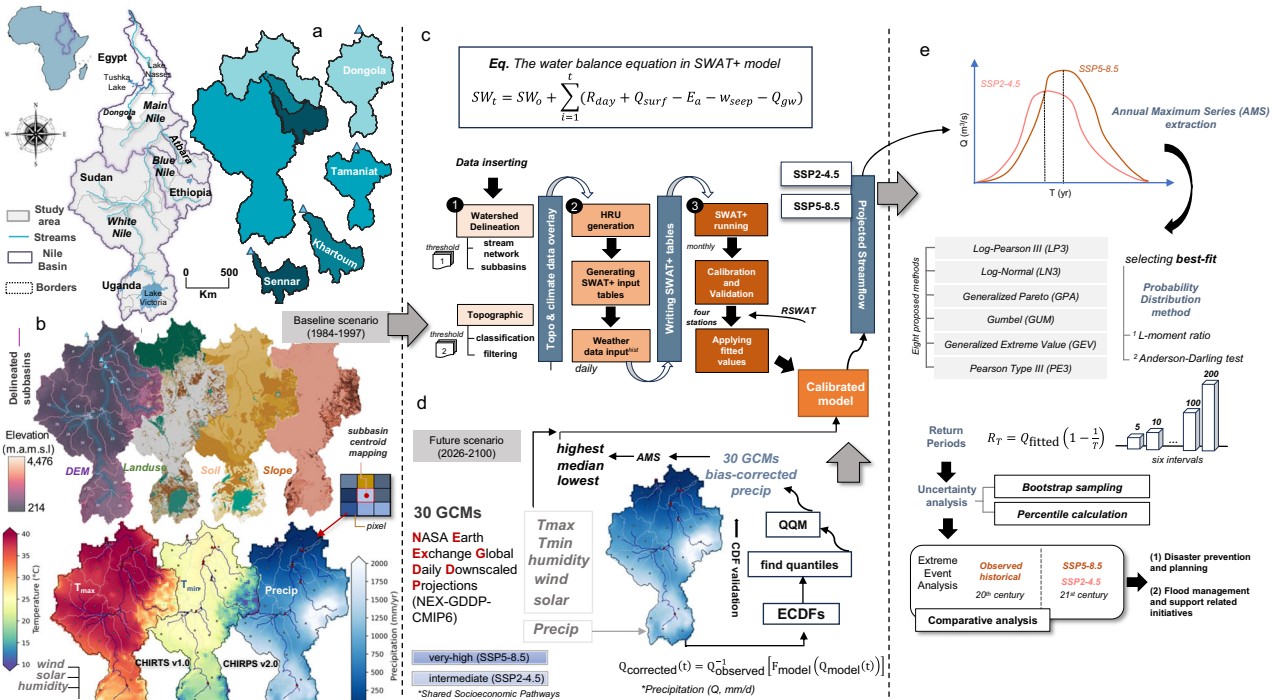

**Fig. 1 | Integrated hydrological framework for assessing climate-driven streamflow extremes in the Nile Basin. a** The Nile Basin and its three main sub-basins: the White Nile, Blue Nile, and the Atbara. Also shown are the four gauging stations (Sennar, Khartoum, Tamaniat, and Dongola) and their respective sub-basins. **b, c** Construction, simulation (using QSWAT+ and SWAT+), calibration, and validation (using RSWAT) of the continuous rainfall-runoff model; digital elevation model (DEM), land use, soil, and slope data were used as model inputs to define watershed, stream networks, and hydrological response units (HRUs), and meteorological inputs (CHIRPS precipitation, CHIRTS temperature, and CFSR

wind speed, solar radiation and relative humidity) were used to simulate runoff. **d** Future climate projections using bias-corrected precipitation from 30 climate models under SSP2-4.5 and SSP5-8.5 scenarios (CMIP6 data). The climate variables projected from three selected models, representing the highest, median, and lowest annual maximum series (AMS) precipitation values for each scenario, were used as inputs for the calibrated SWAT+ model to simulate future streamflow. **e** Extreme event analysis applied to the AMS streamflows (1984–2016 and 2026–2100) to estimate intensity of extreme streamflows at return periods of 5, 10, 25, 50, 100, and 200 years in the 20th (observed) and 21st centuries (SSP2-4.5 and SSP5-8.5).

(GCMs) and projected significant increases in future flood extremes; however, the analysis was limited in spatial coverage and model diversity[12]. Earlier investigations have attributed observed changes in river flow and sediment loads across the Nile Basin to historical climate variability, highlighting the system's sensitivity to both past and future climatic drivers[13]. In the Upper Blue Nile, a regional climate–hydrology framework indicated increases in temperature, decreases in precipitation, and reductions in total water yield by as much as 22.7% under RCP8.5 by the late century[14]. Coupled Model Intercomparison Project Phase 3 (CMIP3) projections suggested precipitation increases of 7–48% and streamflow increases of 21–97%, but emphasized a shift in rainfall seasonality and the potential consequences for hydrology[15]. By contrast, CMIP6 GCMs display large biases in simulating annual precipitation, underestimate interannual variability, and disagree on the sign of projected changes, reinforcing the challenges of using climate model outputs directly in the basin[16]. Other recent work also points to future reductions in streamflow and water yield of up to 54% in the Upper Blue Nile, driven by rising evapotranspiration and declining rainfall[17]. Despite these advances, many studies remain limited to individual sub-basins or rely on small ensembles of climate models, reducing their ability to capture the full range of uncertainty.

Several global assessments have also underscored that the Nile is among the most challenging large-scale basins for hydrological modeling because of its size, contrasting climatic regimes, and data limitations[18–21]. These challenges further highlight the significance of efforts to reproduce streamflows across the basin. To overcome these limitations, we develop a climate-driven hydrological framework by implementing a fully calibrated Soil and Water Assessment Tool (SWAT+) model that covers the entire Nile Basin, with a focus on the downstream countries. Three representative projections drive this model, reflecting the highest, median, and lowest

futures, selected from 30 CMIP6 models in the NASA Earth Exchange Global Daily Downscaled Projections (NEX-GDDP) dataset. By incorporating two Shared Socioeconomic Pathways (SSP2-4.5 and SSP5-8.5), our approach enables a more robust representation of the potential range of future climate conditions. While we recognize the importance of anthropogenic drivers such as dam operations, irrigation, and land-use change, this study focuses on the effects of climate variability to establish a baseline of naturalized hydrological responses. This strategy aligns with best practices in large-basin modeling, especially in transboundary contexts where operational rules and anthropogenic data are often incomplete or inconsistent. We introduce an open-source Python framework that automates climate data processing, SWAT+ integration, and extreme event analysis, thereby enhancing reproducibility and broad applicability. By capturing climate-driven extremes, our workflow supports robust, basin-wide assessments of future flood risks, streamflow variability, and societal vulnerabilities under divergent emissions scenarios.

## Results

We built a SWAT+ model for the Nile Basin and calibrated the model against streamflow data from four stations (Sennar, Khartoum, Tamaniat, and Dongola; Fig. 1b), hereafter referred to as four gauging stations. We used bias-corrected climate projections (NEX-GDDP-CMIP6) as the forcing input for the calibrated SWAT+ model to simulate future streamflows under the SSP2-4.5 and SSP5-8.5 scenarios at the Dongola station (Fig. 1a). An extreme event analysis was conducted to assess flood magnitudes and the frequency of extremes. The uncertainties of extremes resulting from future projections were evaluated using a bootstrap method. The overall framework of the study is illustrated in Fig. 1, with full details provided in the Methods section.

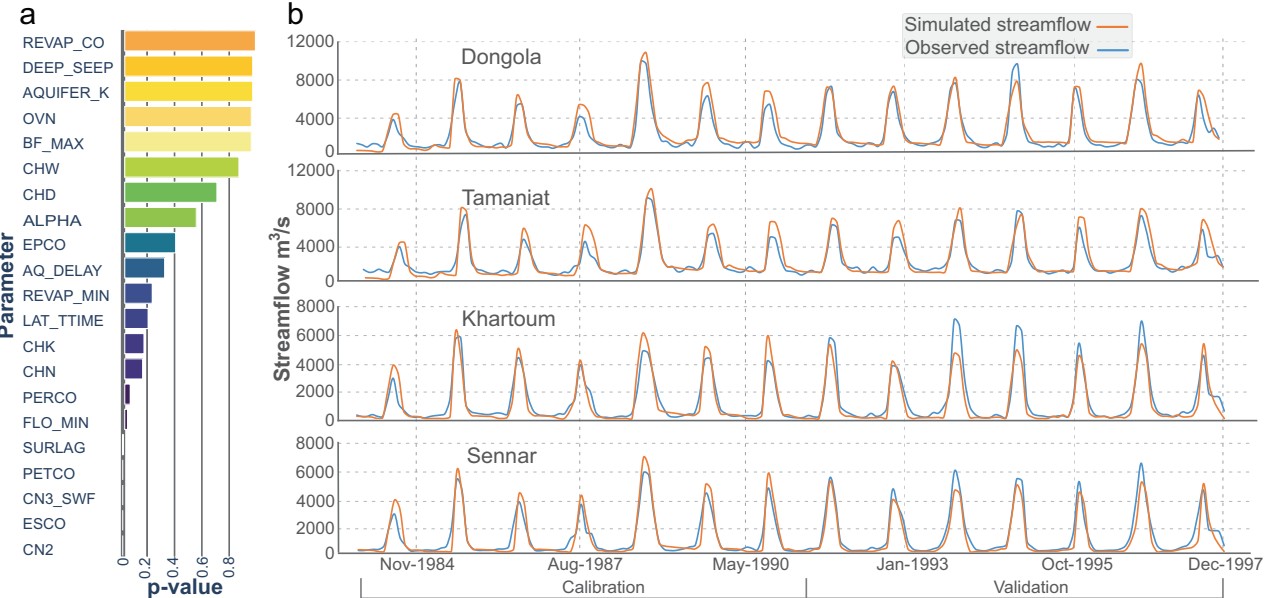

**Fig. 2 | Sensitivity analysis, model calibration and validation of streamflow at key gauging stations. a** The plot displays the model calibration parameters, where lower *p*-values indicate higher sensitivity. **b** Comparison of observed and simulated monthly streamflow at the four gauging stations (Sennar, Khartoum, Tamaniat, and Dongola). The calibration period extends from Jan-1984 to Dec-1991 and the validation period from Jan-1992 to Dec-1997.

## SWAT+ model (construction, simulation, calibration and validation)

The delineation of watersheds, stream networks, and hydrological response units (HRUs) within the QSWAT+ model resulted in 34 subbasins and 234 HRUs, each representing unique combinations of land use, soil, and slope (Fig. 1b). Monthly streamflow values (in m³/s) for each channel in the watershed were generated for the simulation period (1984–1997), including the simulated streamflow at the four key gauging stations. The calibration period spanned from 1984 to 1991, and the validation period from 1992 to 1997. Based on *P*-values, sensitivity analysis identified CN2, ESCO, CN3_SWF, PETCO, and SURLAG as the most influential parameters governing surface runoff, soil moisture retention, and streamflow dynamics (Fig. 2a). The observed monthly streamflow ranged from 9722 m³/s (August 1988) to 459 m³/s (February 1991) in Dongola, from 8904 m³/s (August 1988) to 543 m³/s (February 1987) at the Tamaniat station, from 7089 m³/s (August 1993) to 47 m³/s (February 1985) in Khartoum, and ranged from 6481 m³/s (August 1996) to 48 m³/s (February 1991) at the Sennar station (Fig. 2b). The model's performance in capturing both high- and low-flow conditions is essential for validating its hydrological reliability. A comparison of simulated versus observed monthly streamflow during the calibration period (1984–1991) and the validation period (1992–1997) shows strong model performance at the four gauging stations (Fig. 2b). Detailed performance metrics for key gauging stations are provided in Supplementary Text 2 and Supplementary Table 1.

## Impact of climate change on extreme hydrological events

QQM was applied to the 2015–2024 period, when both modeled and observed precipitation data were available to identify and correct biases in the climate model's precipitation. The derived correction was then applied to the future projections. The bias-correction results demonstrate improvements in the alignment between the cumulative distribution functions (CDFs) of the observed Climate Hazards Group InfraRed Precipitation with Stations (CHIRPS) and the corrected model data under the SSP2-4.5 and SSP5-8.5 scenarios, as applied to the 90th and 99th percentiles (Supplementary Fig. 2). Bias-corrected precipitation data for 2025–2100 were used to derive the annual maximum series (AMS) across 30 models (Fig. 3). Under SSP2-4.5, the aggregated AMS values ranged from 805 mm (lowest: ACCESS-ESM1-5) to 1045 mm (highest: CNRM-ESM2-1), with a

median of 863 mm (MPI-ESM1-2-HR). Similarly, under SSP5-8.5, the aggregated AMS ranged from 825 mm (lowest: ACCESS-ESM1-5) to 1132 mm (highest: CNRM-ESM2-1), with a median of 946 mm (GISS-E2-1-G). When compared to the historical period (1984–2016), which exhibited a mean AMS precipitation of 9.7 mm/day (Supplementary Fig. 3), the representative three-model mean increases to 12.5 mm/day under SSP2-4.5 (~29% higher) and 13.7 mm/day under SSP5-8.5 (~41% higher).

Three GCMs were selected from each scenario (SSP2-4.5 and SSP5-8.5) based on their projected AMS, representing the highest, median, and lowest AMS values. These selected GCMs were then used to drive the SWAT + model for simulating future monthly streamflow (2026–2100). The streamflow predicted under the SSP2-4.5 and SSP5-8.5 scenarios was analyzed with the extreme event method to evaluate the effects of climate change on the intensity and frequency of hydrological extremes. The intensity of streamflow at a return period ranging from 5 to 200 years was derived using the log Pearson Type III Distribution (LP3), the best-fit model for this study. The uncertainty of streamflow extremes was quantified by a range of variations between the 5th and 95th percentiles based on 1000 bootstrap samples drawn from the projected AMS data from the three model outputs of the two scenarios.

Figure 4 presents the return period plots, comparing observed data from the 20th century (1984–2016) with projections for the 21st century (2026–2100) under both scenarios. The historical 50- and 200-year return periods for peak streamflows were estimated at 13,400 and 14,900 m³/s, respectively. Under SSP2-4.5 (50th percentile), these values are projected to rise significantly, with the 50- and 200-year return periods increasing to 21,800 and 24,100 m³/s, respectively. Similarly, SSP5-8.5 (50th percentile) projects an even greater amplification of extreme events, with the 50-year return period exceeding 24,700 m³/s and the 200-year return period exceeding 27,400 m³/s. The differences between the 95th and 5th percentile values express the uncertainties in the projected streamflow extremes. Based on the median (50th percentile) values of streamflows for the 21st century, the intensity of extreme streamflows is expected to increase by 49%, 57%, 62%, 63%, 63%, and 61%, corresponding to 5-, 10-, 25-, 50-, 100-, and 200-year events, respectively, under SSP2-4.5. Similarly, under SSP5-8.5, the streamflow extremes are projected to increase by 73%, 80%, 84%, 85%, 85%, and 84% for 5-, 10-, 25-, 50-, 100-, and 200-year return periods, respectively. These differences highlight the increasing flood risks and variability under

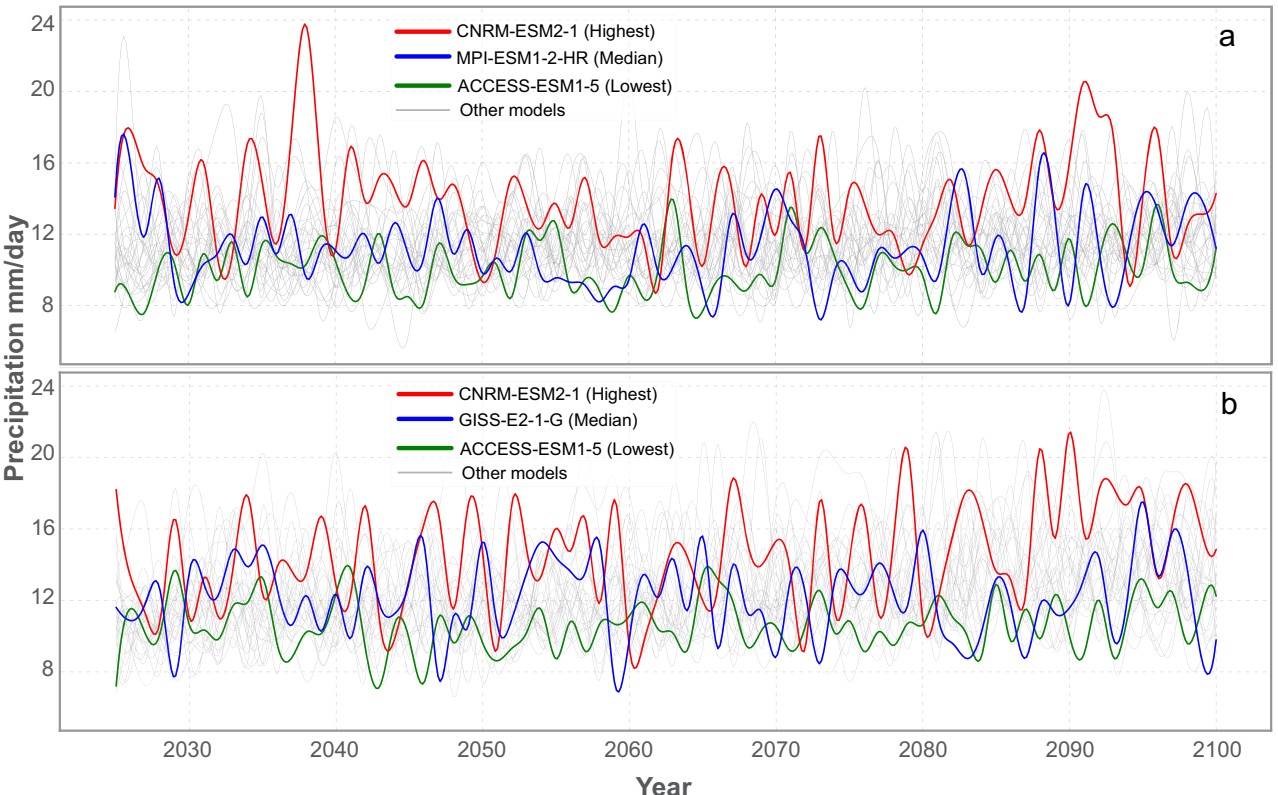

**Fig. 3 | Annual maximum series (AMS) projections for SSP2-4.5 and SSP5-8.5.**
**a** AMS projections for the 30 GCM models under SSP2-4.5 across the study area encompassing the aggregated four subbasins within the Nile Basin; the projections highlight the models with the highest, median, and lowest aggregated values for the period 2025–2100. **b** AMS projections under SSP5-8.5 across the study area, identifying the highest, median, and lowest models based on cumulative AMS values.

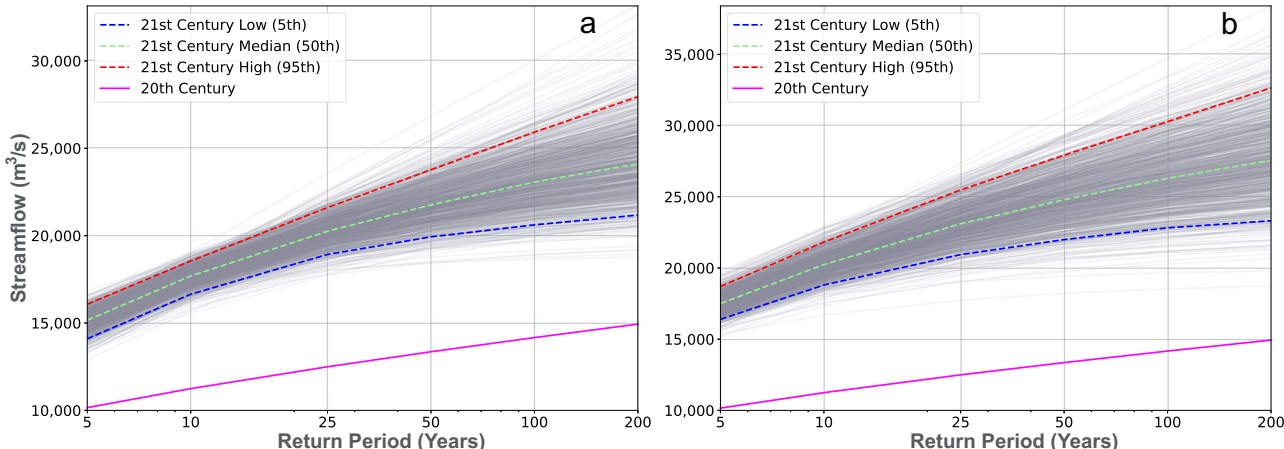

**Fig. 4 | Analysis of extreme streamflow events under SSP2-4.5 and SSP5-8.5 at Dongola station.** Plots of streamflow extremes comparing streamflow intensity from the 20th century observations with projections for the 21st century under higher emissions scenarios, emphasizing the need for robust water management strategies. We also conducted an extreme event analysis of precipitation over the upstream aggregated Nile subbasins that drain at Dongola station to validate the projected increase in streamflow in the 21st century. The analysis reveals an intensification of precipitation extremes during the 21st century, consistent with projected increases in extreme streamflow. These results are provided in Supplementary Fig. 4 and discussed in Supplementary Text 3.

**a** SSP2-4.5 and **b** SSP5-8.5. The predicted extremes include streamflow intensity at return periods of 5, 10, 25, 50, 100, and 200 years, with the uncertainty represented by the 5th and 95th percentiles.

The projected increase in 21st-century streamflow reaching Dongola station aligns with recent GRACE and GRACE-FO observations of terrestrial water storage (TWS) over the Nile Basin, providing independent evidence that the basin has become wetter over the past two decades. TWS, which reflects changes in surface water, soil moisture, and groundwater, increased by 0.24 cm/yr (632 BCM/yr) between 2002 and 2017 and by 2.4 cm/yr (6356 BCM/yr) between 2018 and 2024 (Supplementary Fig. 5). These observed increases are consistent with the intensification of extreme

precipitation events already underway in the basin. While GRACE observations capture only the recent past and cannot be extrapolated to future conditions by themselves, they lend support to the credibility of climate model projections. Ultimately, the projected rise in streamflow extremes across the 21st century is primarily driven by CMIP6-based increases in precipitation and runoff, with GRACE serving as a supporting line of evidence for recent consistency.

## Discussion

The modeling framework developed for the Nile Basin in this study represents a significant advancement in understanding basin-wide hydrology and its implications for downstream nations under historical and future climate scenarios. While earlier studies often focused on sub-basins or limited climate scenarios, our work employs a fully integrated approach that couples the SWAT+ model with 30 bias-corrected CMIP6 models at the basin-wide scale. This framework captures temporal variability across multiple plausible climate scenarios (SSP2-4.5 and SSP5-8.5), providing insights into flood hazards downstream. Moreover, the open-source Python framework developed here automates the entire workflow, covering climate data processing through to extreme event analysis, and provides a reproducible and transferable method. It also establishes a foundation for integrating anthropogenic drivers such as dam operations and for advancing policy-oriented water governance analyses in large river systems worldwide.

Based on the 50th percentile (median) projections, our results show a pronounced intensification of extreme flooding events under both scenarios, with SSP5-8.5 exhibiting more dramatic changes. For observed data, the 50-year return period peak streamflow was approximately 13,400 m³/s. Under the SSP2-4.5 scenario, this value is expected to increase substantially, with the median 50-year return period flow projected to reach approximately 21,800 m³/s. Under SSP5-8.5, the 50-year return period flow increases substantially, exceeding 24,700 m³/s. Similarly, the historical 200-year return period peak streamflow of 14,900 m³/s is projected to rise to 24,100 m³/s under SSP2-4.5 and exceed 27,400 m³/s under SSP5-8.5. Return periods for extreme events are also changing: for example, the historical 100-year flood (~14,200 m³/s) is projected to occur roughly every 4 years under SSP2-4.5, and about every 2.75 years under SSP5-8.5. These central projections fall within a wider 5th to 95th percentile range across the model ensemble, capturing the spectrum from low- to high-impact climate futures. This range allows policymakers and planners to evaluate risks under both moderate and worst-case scenarios, supporting informed decisions on infrastructure design, early warning systems, and adaptive water management strategies such as modular and scalable systems that can withstand future uncertainties. Complementary analyses of precipitation extremes also revealed systematic increases across return periods, consistent with the streamflow results. Such projections are consistent with both observational evidence and earlier modeling efforts from the Nile Basin. For instance, since the construction of the Aswan High Dam and the formation of the Lake Nasser reservoir in the 1970s, the recurring Nile floods have never exceeded the reservoir's storage capacity (150 BCM). However, between 1998 and 2003, recurring extreme Nile floods filled Lake Nasser for the first time, and excess waters amounting to 27 BCM had to be diverted from the Lake. Two decades later, during 2019–2023, even more severe flood events filled the Lake again, prompting the diversion of 53.5 BCM[22].

These observations provide observational evidence of increasing flood intensity and frequency, reinforcing our findings that extreme floods will become more common and severe in the 21st century. In support of this, prior studies[23,24] have shown that 100-year floods from the 20th century could recur every 2–25 years under a warming climate in the Nile Basin. Moreover, regional projections indicate a 152% increase in 50–100-year floods and a 182% increase in 100–200-year floods across Africa by the end of the 21st century[23,24]. These historical patterns and future projections underscore the pressing need for updated flood risk assessments and adaptive strategies. Our projections of intensified flood frequencies and magnitudes align with a growing body of work indicating that global warming accelerates the

hydrological cycle[25]. Studies across multiple continents have documented rising peak flows and more frequent extreme precipitation events[23,26]. In particular, research on large river basins such as the Amazon and the Indus suggests parallel shifts toward more extreme streamflow regimes under high-emission scenarios[27,28]. A significant practical implication of our findings is recognizing extreme floods as a resource rather than merely a hazard[29,30]. For instance, a previous study demonstrated that releasing surplus floodwater (53.5 BCM) from Lake Nasser into high-infiltration depressions south of the Tushka Lakes (Fig. 1a) could recharge the fossil waters of the Nubian Sandstone Aquifer and mitigate flood impacts. Their modeled scenarios, focusing on the period 2019–2023, indicated that up to 75% of these released waters (39.7 BCM) could be stored naturally underground[22]. Our findings suggest that under SSP5-8.5, the magnitude and frequency of extreme floods are likely to increase further, which reinforces the growing importance of proactive water management strategies. Identifying additional recharge settings could help downstream countries integrate flood mitigation with sustainable groundwater development, ultimately turning extreme events into opportunities for long-term water resilience.

Although this study focuses on climate-driven impacts in the Nile Basin's downstream nations, the framework is adaptable for analyzing anthropogenic influences such as dam operations, land-use changes, and evolving water demands. Incorporating reservoir operation strategies, including filling schedules, release rules, and multi-objective trade-offs, would provide critical insight into how infrastructure developments affect water availability, flood risk, and transboundary resource sharing[8]. This is particularly relevant for ongoing developments such as the Grand Ethiopian Renaissance Dam (GERD), which have significant implications for regional hydropower generation, water allocation, and political stability[7]. The framework's scenario-based planning features allow stakeholders to examine various combinations of climatic and human-driven factors, enhancing their ability to anticipate future outcomes more effectively. Similar challenges, including increasing flood intensity, complex reservoir management, and allocation conflicts, are also present in other river basins, such as the Ganges-Brahmaputra[31], Mekong[32], and Tigris-Euphrates[33]. Applying integrated modeling approaches can support conflict-sensitive water governance and long-term adaptation planning in other climate-stressed regions[25,34]. Further enhancements, such as incorporating reservoir operations, land-use changes, evolving water demands, coupling with short-term forecasts, and enabling real-time data-sharing protocols, can expand the framework's utility. Even in its current form, this study provides a replicable and basin-wide benchmark for assessing climate-driven hydrologic extremes. It introduces a reproducible, open-source framework that allows for the future integration of additional drivers. The approach delivers timely and actionable insights for managing downstream flood risks and guiding policy across vulnerable river systems.

## Methods

The methodology involves four main steps (Fig. 1b–e). We constructed a rainfall-runoff model (SWAT+) over the study area (Step I) and calibrated and validated the model against streamflow data (Step II). In step III, we first bias-corrected climate projections from the NEX-GDDP-CMIP6 dataset, and then used them as inputs to the calibrated SWAT+ model to generate future streamflows under the SSP2-4.5 and SSP5-8.5 scenarios. Finally, we applied extreme event analysis to estimate changes in streamflow extremes at various return periods. A parallel assessment of precipitation extremes was also performed, enabling comparisons between projected changes in precipitation and their hydrological manifestation in streamflow. We conducted a bootstrap analysis, sampling three representative model outputs, to quantify the uncertainties of predicted flood risks (Step IV). All data types used in this study and their sources, including spatial and meteorological inputs, are listed in Supplementary Table 2.

### Construction and simulation of a SWAT+ model

SWAT+ is a widely recognized hydrological modeling framework designed to predict the impacts of land use, management practices, and climate

change on water, sediment, and agricultural chemical yields in watersheds or large river basins[35]. The model operates as a semi-distributed, continuous-time model, leveraging Geographic Information Systems (GIS) to subdivide watersheds into smaller units known as HRUs to simulate spatial variability effectively. We used the QSWAT + 3.0 plugin within QGIS to delineate the watershed and stream networks of the Nile Basin, utilizing the Shuttle Radar Topography Mission Digital Elevation Model (SRTM 90 m DEM; Fig. 1b). Four gauging stations were specified as sub-basin outlets, where each of the delineated subbasins (Fig. 1a, b) is characterized by unique topographical and hydrological attributes essential for subsequent HRU analysis and hydrological simulations. HRUs were delineated using three key spatial datasets: land use, soil type, and slope. Unique combinations of these datasets define each HRU, allowing the model to simulate spatial variability in key hydrological processes (e.g., infiltration, evapotranspiration, surface runoff)[36]. Land use data (e.g., forests, agricultural areas, and urban zones) were obtained from the USGS Global Land Cover Characterization (GLCC) dataset, soil data from the FAO Global Soil Map, and the slope from the SRTM 90 m DEM (Fig. 1b). Thresholds of 10% for land use, soil type, and slope were applied during HRU generation to ensure computational efficiency without compromising the model's accuracy.

The simulation of hydrological processes within the Nile Basin was conducted using SWAT+ Editor 3.0.8. with daily precipitation and temperature serving as the primary climate inputs. Precipitation data were extracted from the CHIRPS dataset, and temperature data from the Climate Hazards Center InfraRed Temperature with Stations (CHIRTS) dataset. CHIRPS and CHIRTS integrate satellite imagery with ground-based station data (Fig. 1b). CHIRPS delivers high-resolution gridded precipitation data at a 0.25° spatial resolution, while CHIRTS provides consistent temperature records, both of which are essential for accurately simulating evapotranspiration[37,38].

Three additional auxiliary meteorological parameters (relative humidity, wind speed, and solar radiation) were extracted from the Climate Forecast System Reanalysis (CFSR) global dataset and included in the simulations. The CFSR dataset integrates satellite observations with global atmospheric models to produce reanalysis data at a 0.5° spatial resolution[39]. The simulation period covered 1984–1997, during which streamflow data from the four gauging stations were available, with the first year designated as a warm-up period.

During the simulation, SWAT+ solved water balance equations iteratively across HRUs, sub-basins, and the entire watershed, integrating spatial and temporal variability. Key hydrological processes were modeled. Potential evapotranspiration (PET) was estimated using the Penman-Monteith method, initial abstraction was calculated using the Curve Number method, and water routing within channels was performed using the variable storage method. The primary outputs were monthly streamflow data, which were used for subsequent calibration and validation against observed data from key gauging stations.

## Calibration and validation of the SWAT+ model

We calibrated the SWAT+ model for the Nile Basin using RSWAT version 4.01—an open-source software package within the R environment, designed for hydrological parameter calibration, sensitivity analysis, and uncertainty assessment[40]. RSWAT supports parallel processing and multi-site calibration, and employs various algorithms to refine parameter values efficiently. We used the Latin hypercube sampling (LHS) method (a built-in function in RSWAT) to generate a wide range of parameter combinations for calibration and sensitivity analysis. LHS is a stratified random sampling technique that efficiently explores the full range of each parameter, supporting the identification of optimal sets that maximize the Nash–Sutcliffe efficiency (NSE) during calibration. The "sensitivity" functions available in RSWAT[41] facilitated the identification of key parameters that largely control the basin's hydrological response.

The calibration targeted four gauging stations where monthly streamflow records were reported by the Egyptian Ministry of Water

Resources and Irrigation[42]. The dataset was split into a calibration period from January 1984 to December 1991 and a validation period from January 1992 to December 1997. The model calibration was guided using the NSE function. The calibration process involved 300–500 iterations, combining automated calibration at multiple gauging stations with manual fine-tuning. This approach enabled the model to match observed streamflow patterns and account for hydrological differences across various regions of the Nile Basin[43]. Hydrological parameters were selected during the calibration based on their sensitivity and influence on model performance. These parameters are saved in the *calibration.cal* file, provided as supplementary material. It can be loaded into SWAT+ Editor, enabling efficient reproduction and application of the calibrated model for further analysis. The overall SWAT+ model performance for the calibration and validation periods was evaluated using multiple statistical metrics, including the NSE function, percent bias (PBIAS), and Kling–Gupta efficiency (KGE). The results are presented in Supplementary Text 2 and Supplementary Table 1.

## Future climate projections and bias correction

The projected climatic data used in this study were obtained from the NEX-GDDP-CMIP6 dataset. It is a high-resolution dataset that integrates projections from multiple GCMs under varying scenarios[44]. NEX-GDDP downscales CMIP6 outputs to a spatial resolution of $0.25° \times 0.25°$ using advanced bias-correction and downscaling techniques[45]. The dataset includes projections for multiple SSPs. This study focused on SSP2-4.5, a stabilization pathway with intermediate mitigation efforts, and SSP5-8.5, which assumes high emissions and a fossil-fuel-intensive future[46].

From the NEX-GDDP dataset, we selected 30 GCMs that provide the complete set of required variables (precipitation, maximum and minimum temperature, relative humidity, solar radiation, and wind speed) to run SWAT+ model across the Nile Basin for both SSP2-4.5 and SSP5-8.5 scenarios. Daily observed historical parameters were incorporated for bias correction and model validation, sourced from CHIRPS for 2015 to 2024. CHIRPS was selected as the observational reference because of its high spatial resolution, long-term coverage (1981–present), and widespread validation across Africa, where it has generally outperformed other satellite–gauge products in hydroclimatic applications[37,47]. Numerous studies across the Nile Basin and East Africa have confirmed that CHIRPS reliably captures rainfall variability, with strong correlations to gauge data[48,49]. While performance may vary in regions with sparse gauges or complex terrain, and occasional biases have been reported for extreme events, CHIRPS remains one of the most reliable and extensively used precipitation datasets for Africa. Its adoption ensures a consistent and validated baseline for bias correction and hydrological modelling in the Nile Basin. The downloaded data were preprocessed to ensure consistency across time formats and missing data points. We applied QQM to correct systematic biases in climate model outputs by aligning their distributions with observed CHIRPS data over the 2015–2024 overlap period.

Although NEX-GDDP-CMIP6 already provides bias-corrected variables, residual discrepancies remain between model outputs and regional observations, particularly in the upper percentiles of precipitation extremes (Supplementary Fig. 2). We therefore applied an additional quantile–quantile mapping step to ensure consistency with CHIRPS, which also served as the baseline dataset for SWAT+ calibration. This approach follows other regional studies that emphasize the need for localized corrections to capture precipitation extremes more accurately[12,30,50]. The model data were divided into 100 percentiles, allowing precise adjustment across the full distribution to better match observed variability. The transformation is expressed mathematically in Eq. (1) as

$$Q_{corrected}(t) = Q_{observed}^{-1}[F_{model}(Q_{model}(t))] \qquad (1)$$

where $Q_{corrected}(t)$ is the bias-corrected value at time $t$, $Q_{observed}$ is the inverse of the CDF of observed data, $F_{model}$ is the CDF of modeled data, $Q_{model}(t)$ is the modeled value at time $t$. This method preserves the temporal structure of

the data while ensuring that statistical properties, such as the mean, variance, and skewness, closely align with the observations. The percentile-based adjustment accurately captures both extreme values and central tendencies in the data distributions, thereby improving the effectiveness of the bias correction[51,52]. Validation was performed by plotting the CDFs for observed and model data before and after bias correction, focusing on the 90th and 99th percentiles (Supplementary Fig. 2). Bias correction was exclusively applied to precipitation, as it is the primary driver of streamflow and exhibits significant biases in GCM outputs compared to observed data.

In contrast, temperature data from the climate models showed strong correspondence with observed CHIRTS data during the overlapping period of 2015–2016 across 30 climate models, with a minimal shift ranging from ±1 to 1.5 °C. This consistency negated the need for temperature bias correction. This approach aligns with a recent study where precipitation exhibited the most significant variability and discrepancies among climatic parameters, and applied bias correction only to precipitation over the Blue Nile Basin[12]. Hydrological modeling studies elsewhere recognize precipitation's dominant role in streamflow variability[14,53]. Once bias corrections were established for the overlap period, the adjustments were applied to the future projections of the 30 climatic models for 2025–2100.

To account for the range of uncertainty represented by the 30 climate models, we selected three representative models for each scenario (SSP2-4.5 and SSP5-8.5) based on their AMS values computed across the entire study area. From this set of 30 models, those with the highest, median, and lowest AMS values were chosen. For each selected model, five daily climate variables (precipitation, maximum and minimum temperature, wind speed, relative humidity, and solar radiation) were extracted and used to drive the calibrated SWAT+ model to simulate streamflow throughout the 21st century (2026–2100, with 2025 as the warm-up year). The SWAT+ model had previously been calibrated using historical streamflow data from 1984 to 1991 and validated over the period 1992 to 1997. It was then run through 2016 to extend the baseline simulation under historical (20th-century) climate conditions. This historical baseline (1984–2016) served as a reference for comparing projected streamflow at the Dongola station under future climate scenarios (2026–2100) for both SSP2-4.5 and SSP5-8.5.

### Extreme events analysis

This study employed extreme value analysis to estimate the probability and magnitude of streamflow extremes, with precipitation analyzed in parallel to provide complementary evidence and ensure consistency between climatic drivers and hydrological responses. A block-maxima approach was used to extract AMS from both variables and fit them to statistical distributions for analyzing the tail behavior of streamflows. Six probability distributions were tested: generalized extreme value (GEV), generalized Pareto (GPA), Gumbel distribution (GUM), 3-parameter lognormal (LN3), log Pearson type III (LP3), and Pearson type III (PE3). Distributions were fitted to AMS data using the L-moment method, a robust technique that reduces biases in skewed datasets[54]. The best distribution was selected based on the ratio of L-moment parameters 3 (kurtosis) to 4 (skewness)[55] and Anderson–Darling test results (p-value and test statistics)[56]. More details are provided in Supplementary Text 4. The ranking scores for each of the six distributions were calculated (Supplementary Table 3 for streamflow; Supplementary Table 4 for precipitation). The top three distributions showed similar behavior for both variables (Supplementary Figs. 6 and 7). The distribution LP3 had the best-fit score and was selected for extreme event analysis.

The extremes at return periods of 5, 10, 25, 50, 100, and 200 years ($R_T$) were calculated in Eq. (2) as follows:

$$R_T = Q_{\text{fitted}}\left(1 - \frac{1}{T}\right) \qquad (2)$$

where $T$ is the return period and $Q_{\text{fitted}}$ is the inverse CDF of the best-fit probability distribution, which provides the variable value corresponding to the exceedance probability $\left(1 - \frac{1}{T}\right)$[57]. By estimating the intensity of the variable for varying return periods, the analysis highlights changes in the magnitude of extreme events between the 20th and 21st centuries under different emission scenarios.

### Uncertainty analysis

The predictions of extremes are affected by forcing data projected by variations of CMIP6 model outputs. To quantify the model uncertainty, a bootstrap method that randomly sampled AMS data from each of three representative models was used to generate an ensemble of 1000 AMS dataset[58,59] for each scenario. This approach assumes that all models are equally valid. The selected probability distribution was fitted to each resampled dataset, and the 5th, 50th (median), and 95th percentiles were calculated[12]. These percentiles defined the uncertainty bounds for the intensity of extremes at each return period.

### Reporting summary

Further information on research design is available in the Nature Portfolio Reporting Summary linked to this article.

### Data availability

The datasets supporting this study are available on Figshare: SWAT calibration file (calibration.cal) can be downloaded using 10.6084/m9.figshare.28319603, and observed streamflow data can be downloaded using 10.6084/m9.figshare.28319618. All other datasets used in this study are listed in Supplementary Table 2, along with their respective sources and access details. The full list of the selected 30 models is presented in Supplementary Table 5.

### Code availability

The custom code used to process and analyze CMIP6 climate data for bias correction and hydrological modeling is openly available on GitHub at https://github.com/heshamgeo/CMIP6-BiasCorrection-SWAT. The repository contains notebooks for data retrieval, preprocessing, Quantile–Quantile Mapping bias correction, SWAT+ input formatting, and extreme event analysis.

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

## Acknowledgements

Funding to Western Michigan University was provided by the National Aeronautics and Space Administration (NASA) Earth Science Division grant (80NSSC244K1155).

## Author contributions

H.E. and M.S. coordinated and wrote the article. H.E. and D.T. constructed and calibrated the SWAT+ model. M.S. and E.Y. supervised the construction and calibration of the SWAT+ models. H.E., M.S., and E.Y. reviewed the article. H.E. and D.T. generated figures. H.E., H.E.T.U., and H.K. conducted statistical analysis and calculated the uncertainties. All authors read and approved the final manuscript. H.E. designed and formulated the codes and GitHub repository.

## Competing interests

The authors declare no competing interests.
