## [Transparent Peer Review file · Communications Earth & Environment]

Nile Basin Flow Regimes Under 21st Century Climate Variability

Corresponding Author: Professor Mohamed Sultan

Version 0:

Decision Letter:

Dear Professor Sultan,

Your manuscript titled "Nile Basin Flow Regimes Under 21st Century Climate Variability" has now been seen by 2 reviewers, whose comments are appended below. In the light of their advice we regret to inform you that we cannot publish your manuscript in *Communications Earth & Environment*.

You will see that the reviewers raise substantive concerns. Taking these points together with our editorial considerations, we are unable to conclude that your manuscript represents a sufficiently novel and compelling advance over the body of related work in the literature. Unfortunately, these reservations are sufficiently important to preclude publication of this study in *Communications Earth & Environment*.

Although we cannot offer to publish your manuscript, I will send a revised version back to the relevant reviewers if you transfer it to <https://www.nature.com/commsustain> *Communications Sustainability* (a new journal expected to start publishing later in 2025), provided you can:

1. Broaden the study's scope to incorporate a wider range of geographic, climatic, and socio-economic factors affecting the Nile Basin.
2. Update model calibration using the most recent hydrological and climate data to improve accuracy and relevance.
3. Document all model assumptions and parameter choices clearly, and provide a thorough review of prior climate change studies relevant to the Nile for context and comparison.

To transfer your manuscript, please use our manuscript transfer portal. For more information, please see our [manuscript transfer FAQ](http://www.nature.com/authors/author_resources/transfer_manuscripts.html?WT.mc_id=EMI_NPG_1511_AUTHORTRANSF&WT.ec_id=AUTHOR) page.

We are sorry that we cannot be more positive on this occasion and thank you for the opportunity to consider your work.

Best regards,

Alireza Bahadori, PhD
Associate Editor
Communications Earth & Environment
Consulting Editor
Communications Sustainability

Reviewers' comments:

Reviewer #1 (Remarks to the Author):

Review

The study presents a comprehensive, basin-wide assessment of the impacts of 21st-century climate variability on extreme streamflow in the Nile Basin. Using the SWAT+ model forced by bias-corrected CMIP6 GCM outputs under SSP2-4.5 and SSP5-8.5 scenarios, the authors: Predict 63–85% increases in 100-year peak discharges, show that extreme flood events will become more frequent, possibly occurring every decade and emphasize the need for international cooperation in transboundary water governance and promote the use of floods for groundwater recharge opportunities. This paper provides information of great importance to policy makers and water managers in the Nile Basin. I recommend that this manuscript be accepted for publication after some minor comments are addressed.

In some locations, the subfigure labels (e.g., a, b) are bolded, while in others they are not. Please unify the formatting of all figure notations for consistency throughout the manuscript.

The following references appear incomplete and may need full publication details (publisher, editor, ISBN, etc.):

Said, R. *The River Nile: Geology, Hydrology and Utilization* (Elsevier, 2013).

Elsaeed, G. Effects of climate change on Egypt's water supply. In *National Security and Human Health Implications of Climate Change* 337–347 (2012).

Sentence Clarification (Lines 173–175): Our findings align with previous studies that suggested that 20th-century 100-year floods may recur every 2–25 years in a warming climate over the Nile Basin and reported that Africa could see a 152% increase in 50–100-year floods and a 182% increase in 100–200-year floods by the century's end.

Suggested revision: Our findings align with previous studies that indicate 100-year floods from the 20th century could recur every 2 to 25 years under a warming climate in the Nile Basin. These studies also project a 152% increase in 50–100-year floods and a 182% increase in 100–200-year floods across Africa by the end of the 21st century.

Simplify and prioritize the most policy-relevant takeaways in the abstract. End with a strong concluding sentence that highlights the novelty (e.g., "This is the first basin-wide projection study of extreme flows using CMIP6 for the Nile Basin").

Some sections in methods are too technical, Please consider simplifying the language where possible, particularly in sections related to model calibration, bias correction. Define all abbreviations at their first use.

What does the 5th–95th percentile range mean for policy or flood infrastructure planning?

Reviewer #2 (Remarks to the Author):

I am very in favour of a welcoming a paper on the future of the Nile, being such an important transboundary river, but the submitted manuscript does not fulfill the expectations for the following reasons:

- 1) The manuscript only addresses climate impacts and does not consider other drivers, such as reservoir developments, irrigation developments and other land use changes that are so important in the region.
- 2) The manuscript calibrates for a period in the 90-ties and does not account for all changes of the past 30 years.
- 3) The manuscript does not provide enough information on the set up of the model. It does not provide how it represented agriculture, irrigation, reservoir operations. It does not provide parameter values. The study is not reproducible.
- 4) The manuscript does not provide a clear overview of past studies on climate change for the Nile.

Full article: Future hydrology and climate in the River Nile basin: a review

For that reason, my recommendation is to reject the manuscript.

Version 1:

Decision Letter:

Dear Professor Sultan,

Your revised manuscript titled "Nile Basin Flow Regimes Under 21st Century Climate Variability" has now been seen by our original reviewer 1 and a new reviewer 3, who replaces the original reviewer 2 (reviewer 2 was not available for further comments). All comments appear below. In light of their advice we are delighted to say that we are happy, in principle, to

publish a suitably revised version in Communications Earth & Environment, provided you strengthen the justification for streamflow projections by providing clear evidence of model performance across varying hydrological conditions, and fully clarify the limitations of using GRACE data.

We therefore invite you to revise your paper one last time to address the remaining concerns of our reviewer 3. At the same time we ask that you edit your manuscript to comply with our format requirements and to maximise the accessibility and therefore the impact of your work.

EDITORIAL REQUESTS:

****Please take care to match our formatting and policy requirements. We will check revised manuscript and return manuscripts that do not comply. Such requests will lead to delays. ****

SUBMISSION INFORMATION:

OPEN ACCESS:

Communications Earth & Environment is a fully open access journal. Articles are made freely accessible on publication. For further information about article processing charges, open access funding, and advice and support from Nature Portfolio, please visit <https://www.nature.com/commsenv/open-access>

Link Redacted

Best regards,

Alireza Bahadori, PhD
Associate Editor
Communications Earth & Environment
Consulting Editor
Communications Sustainability

REVIEWERS' COMMENTS:

Reviewer #1 (Remarks to the Author):

I've carefully reviewed both the authors' detailed responses and the revised manuscript, and I am very pleased with the revisions. The authors have thoughtfully and thoroughly addressed all of my comments and suggestions. In particular:

- They adopted the suggested language edits, improving the manuscript's clarity and readability.
- They revised the methodology section to enhance transparency and reproducibility.
- They improved the reference formatting and ensured completeness.
- They strengthened the policy relevance of the abstract and discussion, particularly regarding uncertainty and planning implications.

In conclusion, the authors have satisfactorily addressed all of my comments and suggestions. Their methodology is both

sound and reproducible, and they have made a commendable effort to share their data and methods publicly via GitHub. They carefully selected reliable and readily accessible streamflow data to calibrate their model, deliberately avoiding the use of unpublished or questionable data. As a result, they present the first credible climate-driven, calibrated model of the Nile Basin, with a particular focus on the downstream countries. Their climate projections are robust, grounded in a well-calibrated SWAT model and informed by three GCMs that capture the variability across approximately 30 models.

Reviewer #3 (Remarks to the Author):

General comment:

This study applies a calibrated, climate-driven SWAT model forced with bias-corrected CMIP6 simulations to assess future flood risk in the Nile Basin, representing the first basin-wide application of its kind. Because the Nile is one of the most challenging large-scale basins for hydrological modelling, this study represents a clear advance and underscores the need to anticipate future extremes for water security and planning. The manuscript is generally well structured and the methodological approach is robust, and I see mostly minor issues that need clarification. My main concern, however, is the relatively weak explanation of the very large projected increases in future streamflow. Additional analysis of how strongly precipitation and other drivers are projected to change would provide evidence to strengthen the conclusions about streamflow projections. Addressing the following comments would, in my view, make the manuscript suitable for publication.

Specific comments:

1. L36: The statement needs a supporting reference.
2. L55-72: Consider updating references to include more recent studies on climate and hydrological projections for the Nile Basin, or global studies that also discuss the Nile.
3. L73-86: Adding a brief note in the introduction that several studies (e.g., Dai and Trenberth 2002; Clark et al. 2015; Ghiggi et al. 2019; Müller et al. 2021) have shown the Nile to be among the most challenging large-scale basins for hydrological modeling would highlight the significance of your effort in reproducing Nile streamflows.

Dai and Trenberth (2002). Estimates of freshwater discharge from continents: Latitudinal and seasonal variations. *JHM*, 3, 660–687.
Clark et al. (2015). Continental runoff into the oceans (1950–2008). *JHM*, 16, 1502–1520.
Ghiggi et al. (2019). GRUN: An observation-based global gridded runoff dataset from 1902 to 2014. *ESSD*, 11, 1655–1674.
Müller et al. (2021). Does the HadGEM3-GC3.1 GCM Overestimate Land Precipitation at High Resolution? A Constraint Based on Observed River Discharge. *JHM*, 22, 2131–2151.
4. Fig. S1: The colorbar ([0.18–2780] mm/year) does not match the caption (0.018–2780 mm/year). Also, 0.18 mm/year seems unrealistically low. Please double-check these values.
5. L125: Please check the text “QQM effectively identified biases in precipitation projections...”. My understanding is that QQM is first applied to the baseline period, not directly to the projections. Do you mean biases in the baseline precipitation, which are then transferred to the projections?
6. Fig. 3, Fig. 4, and related text: It would be useful to also show precipitation differences between the baseline and projection periods, to better explain the large streamflow increases observed. In particular, I wonder how much precipitation actually increases in the projections to justify streamflow rises of 49% or more.
7. Fig. 3: Please specify which catchment the data corresponds to.
8. Fig. 4: Please specify which station the data corresponds to; is it Dongola? I also suggest showing similar plots for all stations. This would allow readers to infer whether the expected extremes originate from a specific sub-basin.
9. L163-172: This argument is somewhat weak. The link between the GRACE-observed TWS increase (2003–2023) and the projected 21st-century streamflow rise is not fully convincing. GRACE provides valuable recent observations but does not ensure continuation of the trend. I recommend emphasizing climate model projections of precipitation and runoff as the primary drivers of future streamflow, with GRACE used mainly as supporting evidence for recent consistency rather than as a predictor.
10. L209: Which studies?
11. L227: Avoid single-sentence paragraphs.
12. L327-338: Given that NEX-GDDP-CMIP6 already provides bias-corrected variables, could you clarify why an additional bias correction was applied to precipitation? Since CHIRPS is satellite-based and there is sparse in-situ coverage in much

of the Nile basin, it would be helpful to discuss potential limitations of this “reference” dataset.

13. L334: Please clarify the criteria used to select the 30 GCMs from NEX-GDDP? Also, provide a list of the selected GCMs as supplementary information.

** Visit Nature Portfolio's author and referees' website at www.nature.com/authors for information about policies, services and author benefits**

Reviewer Comments (black font) and our Response (blue font)

The study presents a comprehensive, basin-wide assessment of the impacts of 21st-century climate variability on extreme streamflow in the Nile Basin. Using the SWAT+ model forced by bias-corrected CMIP6 GCM outputs under SSP2-4.5 and SSP5-8.5 scenarios, the authors: Predict 63–85% increases in 100-year peak discharges, show that extreme flood events will become more frequent, possibly occurring every decade and emphasize the need for international cooperation in transboundary water governance and promote the use of floods for groundwater recharge opportunities. This paper provides information of great importance to policymakers and water managers in the Nile Basin. I recommend that this manuscript be accepted for publication after some minor comments are addressed. Reviewer 1 provided a positive evaluation and recommended publishing the manuscript with minor revisions. We thank Reviewer 1 for summarizing our findings, highlighting the significance of our results to policymakers and managers in the Nile Basin, and recommending the publication of our manuscript.

In some locations, the subfigure labels (e.g., a, b) are bolded, while in others they are not. Please unify the formatting of all figure notations for consistency throughout the manuscript. We did. We unified the formatting of all subfigure labels following a non-bolded style in the revised text.

The following references appear incomplete and may need full publication details (publisher, editor, ISBN, etc.):

- Said, R. *The River Nile: Geology, Hydrology and Utilization* (Elsevier, 2013).
- Elsaheed, G. Effects of climate change on Egypt's water supply. In National Security and Human Health Implications of Climate Change 337–347 (2012).

We revised the references to include complete and accurate publication details in line with the journal's formatting style:

- Said, R. *The River Nile: Geology, Hydrology and Utilization*. Pergamon Press, Oxford (1993).
- Elsaheed, G. Effects of climate change on Egypt's water supply. In Fernando, H. J. S., Klaić, Z. & McCulley, J. L. (eds) *National Security and Human Health Implications of Climate Change* 337–347 (Springer, Dordrecht, 2012).

Sentence Clarification (Lines 241–244): Our findings align with previous studies that suggested that 20th-century 100-year floods may recur every 2–25 years in a warming climate over the Nile Basin and reported that Africa could see a 152% increase in 50–100-year floods and a 182% increase in 100–200-year floods by the century's end. Suggested revision: Our findings align with previous studies that indicate 100-year floods from the 20th century could recur every 2 to 25 years under a warming climate in the Nile Basin. These studies also project a 152% increase in 50–100-year floods and an 182% increase in 100–200-year floods across Africa by the end of the 21st century. We adopted the proposed wording in the revised manuscript (refer to lines 209-212 in the revised text).

Simplify and prioritize the most policy-relevant takeaways in the Abstract. We did. We simplified the Abstract, prioritizing policy-relevant outcomes (e.g., emphasize the need for coordinated planning, provide actionable risk information, and a framework for regional cooperation and preparedness) to mitigate future flood risks and address water security challenges (refer to lines 22-24 in the revised text). End with a strong concluding sentence that highlights the novelty (e.g., “This is the first basin-wide projection study of extreme flows using CMIP6 for the Nile Basin”). We provided a clear/strong statement: “Here, we assess future flood risk in downstream countries using a calibrated, climate-driven Soil and Water Assessment Tool (SWAT+) model, forced by bias-corrected CMIP6 models under the SSP2-4.5 and SSP5-8.5 scenarios, marking the first application of its kind targeting the Nile Basin downstream regions. Refer to lines 17 to 20 in the Abstract.

Some sections in methods are too technical, Please consider simplifying the language where possible, particularly in sections related to model calibration, bias correction. We did. We simplified the text in the Methods, particularly in the model calibration, bias correction, and extreme events analysis subsections. Refer to lines 341-344, 349-352, 366-377, and 379-382. Define all abbreviations at their first use. We did. Refer to lines 43, 47, 61, 82-83, and 125.

What does the 5th–95th percentile range mean for policy or flood infrastructure planning?

The 5th–95th percentile range represents the spread of projected outcomes across the climate models. They provide uncertainties associated with the projected climatic variables that could guide policy and infrastructure planning. Rather than relying on a single deterministic projection, this range allows decision-makers to consider both moderate (central trajectory) and high-end flood scenarios (extremes). For example, the upper end (95th percentile) can inform the design of protective critical infrastructure to withstand more severe events, while the median and lower bounds support adaptive planning (e.g., modular, scalable water systems) across a spectrum of possible futures. This aligns with best practices in climate-resilient water infrastructure design and policy formulation under uncertainty. We have added a paragraph to the discussion section (lines 192–199) explaining how the 5th–95th percentile range supports scenario-based planning and flood preparedness for downstream countries.

Reviewer Comments: Reviewer 2

I am very in favor of welcoming a paper on the future of the Nile, being such an important transboundary river, but the submitted manuscript does not fulfill the expectations for the following reasons:

First reason/concern: The manuscript only addresses climate impacts and does not consider other drivers, such as reservoir developments, irrigation developments, and other land use changes that are so important in the region. We appreciate the reviewer’s emphasis on the importance of the multiple drivers affecting the hydrology of the Nile Basin. However, our study was deliberately designed as a climate-only assessment to isolate the specific impacts of climate variability on flood regimes. We stated this objective in the Abstract, Main, and Discussion sections. We calibrated a basin-wide SWAT+ model under naturalized conditions, excluding reservoirs, irrigation

withdrawals, and land use changes (refer to lines 17-20, 87-89, 174-176, and 182-184 in the revised text).

This modeling choice is not a limitation but a scientifically sound and widely accepted approach in Earth system and hydrological research. Numerous peer-reviewed studies in the Nile Basin and globally have adopted similar strategies to establish a robust baseline of climate-driven hydrologic variability before incorporating more complex anthropogenic factors. Examples include Alfieri et al. (2015), who used EURO-CORDEX climate projections with a rainfall-runoff model to estimate future flood hazard in a naturalized European setting; Lane and Kay (2021), who applied UKCP18 climate data to a national hydrologic model to assess flood and drought risks across Great Britain; Tabari (2020), who demonstrated that climate-induced flood risks intensify with water availability using global hydrologic models; Byun et al. (2019), who assessed hydrologic extremes in Midwestern US watersheds under downscaled GCM projections; and Pal et al. (2023), who used WRF-Hydro to simulate streamflow and flooding in the Northeastern US under future climate scenarios. In the Nile Basin, studies such as Beyene et al. (2010), Stahl et al. (2010), Dile et al. (2013), Szcześniak et al. (2017), Mengistu et al. (2021), Müller et al. (2024), and Nkwasa et al. (2024) similarly employed climate-only modeling approaches to evaluate future hydrologic responses in the absence of anthropogenic drivers.

Incorporating anthropogenic influences, such as dam operations, irrigation practices, and land-use changes, would require detailed, consistent, and transparent datasets. These include time series of reservoir release rules, irrigation withdrawals, and spatially resolved projections of future land use. Unfortunately, such data are often fragmented, confidential, non-public, or unavailable across many Nile Basin countries, posing a substantial challenge to robust and reproducible modeling. In such contexts, attempting to simulate human interventions can introduce significant uncertainties and speculative assumptions, undermining the reliability of results. As emphasized by Lane et al. (2022), Turner et al. (2024), and Prestele et al. (2016), the lack of reliable data on anthropogenic drivers can impair model performance to a greater extent than it enhances realism. Furthermore, studies (e.g., Haddeland et al., 2014) have shown that the effects of climate change alone can match or even exceed those of water infrastructure and abstraction in many basins, reinforcing the need first to quantify the climate signal independently.

Moreover, irrigation within the Nile Basin is heavily concentrated in Egypt and Sudan, which together account for approximately 97% of the 5.6 million hectares of irrigated land in the basin (Mutsch et al., 2017). Since this study focuses on the upstream portion of the basin, where irrigated agriculture is limited, the influence of irrigation on streamflow is expected to be minimal. This supports the decision to exclude irrigation representation in the model, focusing instead on climate-induced hydrologic changes.

Our modeling strategy is also consistent with practices in other heavily managed basins. For example, despite their extensive infrastructure in the Colorado, Columbia, and Mississippi River basins, researchers often first simulate streamflow under naturalized conditions to establish a clean climate baseline before layering in operational complexities. Examples include Christensen and Lettenmaier (2007), Zhou et al. (2018), and Ghimire et al. (2023). This two-step approach remains a common and effective strategy in climate impact modeling.

We do not dismiss the importance of human impacts on the Nile Basin. On the contrary, our manuscript explicitly acknowledges these factors (refer to lines 87-89 and 231-236 in the revised text) and cites recent, relevant studies that explore infrastructure-related effects in the region. Those include Wheeler et al. (2020), Heggy et al. (2023), and Abdelmohsen et al. (2024). Moreover, our modeling framework was designed to be modular and adaptable, allowing for future integration of dam operations, irrigation practices, and land use scenarios when credible data and policy pathways become available. Refer to lines 245-251 in the revised text.

In summary, excluding anthropogenic factors is a strategic and scientifically justified approach. It reflects the limitations in data availability and the need to establish a clear, climate-induced reference baseline across this complex and transboundary basin. Our work provides a solid foundation for future studies that may integrate socio-economic and infrastructure influences as more reliable data become accessible.

Second reason/concern: The manuscript calibrates for a period in the 90s and does not account for all changes of the past 30 years. Our model was calibrated and validated using streamflow observations from 1984 to 1997, specifically those from stations with high-quality, published, and continuous gauge data across key locations in the Nile Basin. Streamflow records beyond the late 1990s are often incomplete, unpublished for many reaches, and unavailable at the basin-wide scale. This limitation has been noted in several studies, some of which relied on restricted or inaccessible datasets for specific sub-basins, limiting transparency and reproducibility (e.g., Beyene et al., 2010; Dile et al., 2013; Mengistu et al., 2021; Melesse et al., 2024; Nkwasa et al., 2024; Magoffin et al., 2025). To ensure the transparency and reproducibility of our analysis, we relied exclusively on publicly accessible data.

Third reason/concern: The manuscript does not provide enough information on the setup of the model. It does not provide how it represented agriculture, irrigation, and reservoir operations. It does not provide parameter values. The study is not reproducible.

We respectfully emphasize that the entire modeling framework is fully documented and publicly accessible, enabling independent researchers to reproduce the study. We provided the following:

- All relevant information about land use data (GLCC), soil, slope, and HRU thresholds is in the Methods section and Supplementary Table 2.
- SWAT Calibration Parameters: The full list of calibrated parameters used in the model is available in the uploaded calibration.cal file hosted on Figshare [DOI: 10.6084/m9.figshare.28319603].
- Observed Streamflow Data: The historical discharge data for the four gauging stations used for calibration and validation are also publicly available [DOI: 10.6084/m9.figshare.28319618].
- Climate Forcing and Bias Correction Code: We developed a fully open-source Python repository that automates the downloading, preprocessing, and bias correction of CMIP6 climate data for hydrological modeling. The code and usage instructions are provided via GitHub: <https://github.com/heshamgeo/CMIP6-BiasCorrection-SWAT>. The historical

climate inputs (CHIRPS, CHIRTS, CFSR) and future CMIP6 model outputs are openly accessible datasets.

These resources and outputs collectively make the SWAT+ model setup and simulation pipeline completely transparent and reproducible. Regarding the reviewer's question on agriculture, irrigation, and reservoir operations, we addressed this in detail in our response to "Reason 1". Our modeling framework is fully reproducible, clearly scoped, and aligned with established best practices for basin-scale climate impact assessments focused on hydrological extremes.

Fourth reason/concern: The manuscript does not provide a clear overview of past studies on climate change for the Nile. Full article: Future hydrology and climate in the River Nile basin: a review. We thank the reviewer for this valuable suggestion and for highlighting the review by Di Baldassarre et al. (2011), which offers a foundational synthesis of early climate and hydrology studies in the Nile Basin. In response, we have revised the Introduction to present a more comprehensive and coherent overview of the existing literature, beginning with the key insights from that review and progressing through more recent studies.

Our revised paragraph emphasizes how early modeling efforts, including those by Kim and Kaluarachchi (2009) and Beyene et al. (2010), revealed considerable divergence in future streamflow projections, primarily due to uncertainties in GCM outputs, downscaling methods, and hydrologic model sensitivities. While these studies laid the necessary groundwork, many focused on sub-basin scales, used small ensembles of earlier GCM models, or lacked integrated, basin-wide perspectives. Refer to lines 42 to 78.

We also include recent studies (e.g., Badawy et al., 2024) that applied rainfall-runoff models to sub-regions of the Nile using a limited number of GCMs. These efforts, although valuable, remain constrained in terms of spatial coverage and model representativeness. To address these limitations, our study applies a fully calibrated SWAT+ model across the entire Nile Basin, with a particular emphasis on downstream countries. We utilize three representative GCM projections (maximum, median, minimum) selected from 30 CMIP6 models under two emissions scenarios (SSP2-4.5 and SSP5-8.5). This method captures the full spread of possible future climate outcomes while allowing for manageable hydrologic simulations. The revised text clarifies how this setup improves on previous efforts and enables one of the most robust, basin-wide climate impact assessments to date.

We believe this revised section fully addresses the reviewer's concern and strengthens the manuscript's contextual foundation.

For that reason, my recommendation is to reject the manuscript. We believe we addressed the four reasons/concerns raised by Reviewer 2, upon which he recommended rejecting our manuscript. We hope the Journal's Reviewers and Editorial Board of Nature Communications, Environment, and Earth find our manuscript publishable in its revised format.

References

- Abdelmohsen, K. *et al.* Watching the Grand Ethiopian Renaissance Dam from a distance: implications for sustainable water management of the Nile water. *PNAS Nexus* **3**, 219 (2024).
- Alfieri, L., Burek, P., Feyen, L. & Forzieri, G. Global warming increases the frequency of river floods in Europe. *Hydrol. Earth Syst. Sci.* **19**, 2247–2260 (2015).
- Badawy, A. *et al.* Floods of Egypt’s Nile in the 21st century. *Sci. Rep.* **14**, 27031 (2024).
- Beyene, T., Lettenmaier, D. P. & Kabat, P. Hydrologic impacts of climate change on the Nile River Basin: implications of the 2007 IPCC scenarios. *Clim. Change* **100**, 433–461 (2010).
- Byun, K., Chiu, C.-M. & Hamlet, A. F. Effects of 21st century climate change on seasonal flow regimes and hydrologic extremes over the Midwest and Great Lakes region of the US. *Sci. Total Environ.* **650**, 1261–1277 (2019).
- Christensen, N. S. & Lettenmaier, D. P. A multimodel ensemble approach to assessment of climate change impacts on the hydrology and water resources of the Colorado River Basin. *Hydrol. Earth Syst. Sci.* **11**, 1417–1434 (2007).
- Di Baldassarre, G. *et al.* Future hydrology and climate in the River Nile basin: a review. *Hydrol. Sci. J.* **56**, 199–211 (2011).
- Dile, Y. T., Berndtsson, R. & Setegn, S. G. Hydrological response to climate change for Gilgel Abay River, in the Lake Tana Basin–Upper Blue Nile Basin of Ethiopia. *PLoS One* **8**, e79296 (2013).
- Ghimire, G. R. *et al.* Insights from Dayflow: a historical streamflow reanalysis dataset for the conterminous United States. *Water Resour. Res.* **59**, e2022WR032312 (2023).
- Haddeland, I. *et al.* Global water resources affected by human interventions and climate change. *Proc. Natl Acad. Sci. USA* **111**, 3251–3256 (2014).
- Heggy, E., Ramah, M. & Abotalib, A. Z. Examining the accuracy of using a single short-term historical flow period to assess the Nile’s downstream water deficit from GERD filling: a technical note. *Earth Syst. Environ.* **7**, 723–732 (2023).
- Kim, U. & Kaluarachchi, J. J. Climate change impacts on water resources in the upper Blue Nile River Basin, Ethiopia. *J. Am. Water Resour. Assoc.* **45**, 1361–1378 (2009).
- Lane, R. *et al.* A large-sample investigation into uncertain climate change impacts on high flows across Great Britain. *Hydrol. Earth Syst. Sci.* **26**, 5535–5556 (2022).
- Lane, R. A. & Kay, A. L. Climate change impact on the magnitude and timing of hydrological extremes across Great Britain. *Front. Water* **3**, 684982 (2021).
- Magoffin, R. H. *et al.* Hydrologic decision support in the Nile Basin: creating status products from the GEOGLOWS hydrologic model. *Hydrology* **12**, 3 (2025).
- Melesse, M. B. & Demissie, Y. Hydrology and droughts in the Nile: a review of key findings and implications. *Water* **16**, 2521 (2024).
- Mengistu, D., Bewket, W., Dosio, A. & Panitz, H. J. Climate change impacts on water resources in the Upper Blue Nile (Abay) River Basin, Ethiopia. *J. Hydrol.* **592**, 125614 (2021).
- Multsch, S., Elshamy, M. E., Batarseh, S., Seid, A. H., Frede, H.-G. & Breuer, L. Improving irrigation efficiency will be insufficient to meet future water demand in the Nile Basin. *J. Hydrol. Reg. Stud.* **12**, 315–330 (2017).

- Müller, O. V. *et al.* River flow in the near future: a global perspective in the context of a high-emission climate change scenario. *Hydrol. Earth Syst. Sci.* **28**, 2179–2197 (2024).
- Nkwasa, A. *et al.* Historical climate impact attribution of changes in river flow and sediment loads in the Nile Basin. *Clim. Change* **177**, 42 (2024).
- Pal, S., Wang, J., Feinstein, J., Yan, E. & Kotamarthi, V. R. Projected changes in extreme streamflow and inland flooding in the mid-21st century over Northeastern United States using ensemble WRF-Hydro simulations. *J. Hydrol. Reg. Stud.* **47**, 101371 (2023).
- Prestele, R. *et al.* Hotspots of uncertainty in land-use and land-cover change projections: a global-scale model comparison. *Glob. Change Biol.* **22**, 3967–3983 (2016).
- Stahl, K. *et al.* Streamflow trends in Europe: evidence from a dataset of near-natural catchments. *Hydrol. Earth Syst. Sci.* **14**, 2367–2382 (2010).
- Szcześniak, M. *et al.* Effect of climate change on hydrology, sediment and nutrient losses in two lowland catchments in Poland. *Water* **9**, 156 (2017).
- Tabari, H. Climate change impact on flood and extreme precipitation increases with water availability. *Sci. Rep.* **10**, 13768 (2020).
- Turner, S. *et al.* Developing water supply reservoir operating rules for large-scale hydrological modelling. *Hydrol. Earth Syst. Sci.* **28**, 4203–4225 (2024).
- Wheeler, K. G., Jeuland, M., Hall, J. W., Zagana, E. & Whittington, D. Understanding and managing new risks on the Nile with the Grand Ethiopian Renaissance Dam. *Nat. Commun.* **11**, 522 (2020).
- Zhou, T. *et al.* Sensitivity of regulated flow regimes to climate change in the western United States. *J. Hydrometeorol.* **19**, 499–515 (2018).

Reviewer Comments (black font) and Our Response (blue font); line numbering refers to the track change version

Reviewer Comments: Reviewer 1

I've carefully reviewed both the authors' detailed responses and the revised manuscript, and I am very pleased with the revisions. The authors have thoughtfully and thoroughly addressed all of my comments and suggestions. In particular:

- They adopted the suggested language edits, improving the manuscript's clarity and readability.
- They revised the methodology section to enhance transparency and reproducibility.
- They improved the reference formatting and ensured completeness.
- They strengthened the policy relevance of the abstract and discussion, particularly regarding uncertainty and planning implications.

In conclusion, the authors have satisfactorily addressed all of my comments and suggestions. Their methodology is both sound and reproducible, and they have made a commendable effort to share their data and methods publicly via GitHub. They carefully selected reliable and readily accessible streamflow data to calibrate their model, deliberately avoiding the use of unpublished or questionable data. As a result, they present the first credible climate-driven, calibrated model of the Nile Basin, with a particular focus on the downstream countries. Their climate projections are robust, grounded in a well-calibrated SWAT model and informed by three GCMs that capture the variability across approximately 30 models.

We sincerely thank Reviewer 1 for his/her thorough review and supportive assessment of our work. We appreciate their recognition of our revisions, methodological rigor, and effort to ensure transparency and reproducibility.

Reviewer Comments: Reviewer 3

General comment: This study applies a calibrated, climate-driven SWAT model forced with bias-corrected CMIP6 simulations to assess future flood risk in the Nile Basin, representing the first basin-wide application of its kind. Because the Nile is one of the most challenging large-scale basins for hydrological modelling, this study represents a clear advance and underscores the need to anticipate future extremes for water security and planning. The manuscript is generally well structured and the methodological approach is robust, and I see mostly minor issues that need clarification. My main concern, however, is the relatively weak explanation of the very large projected increases in future streamflow. Additional analysis of how strongly precipitation and other drivers are projected to change would provide evidence to strengthen the conclusions about streamflow projections. Addressing the following comments would, in my view, make the manuscript suitable for publication. We thank Reviewer 3 for his/her careful evaluation of our manuscript, constructive and insightful comments, and the recognition of our contribution given the challenges of modeling the Nile Basin at a basin-wide scale. The feedback provided by Reviewer 3 helped us improve the clarity and robustness of the paper. We responded to the Reviewer's request to demonstrate that a corresponding increase in precipitation supports the projected increase in future streamflow. In our Response to Comment 6, we added

analyses of precipitation extremes alongside the streamflow analysis. We also addressed each of the issues/concerns raised by Reviewer 3 and revised the manuscript accordingly, as detailed in our Response below.

Reviewer 3 Comment 1 (L36): The statement needs a supporting reference. We provided a supporting reference. Refer to line 36 in the revised text.

Reviewer 3 Comment 2 (L55–72): Consider updating references to include more recent studies on climate and hydrological projections for the Nile Basin, or global studies that also discuss the Nile. We did. We revised the section (now lines 55–74) to replace the earlier references with more recent and basin-relevant studies. Specifically, we incorporated new references such as Badawy et al. (2024), Nkwasa et al. (2024), Mengistu et al. (2021), Roth et al. (2018), Ombadi et al. (2021), and Takele et al. (2022), which strengthen the discussion on recent advances and uncertainties in climate and hydrological projections for the Nile Basin.

Reviewer 3 Comment 3 (L73–86): Adding a brief note in the introduction that several studies (e.g., Dai and Trenberth 2002; Clark et al. 2015; Ghiggi et al. 2019; Müller et al. 2021) have shown the Nile to be among the most challenging large-scale basins for hydrological modeling would highlight the significance of your effort in reproducing Nile streamflows. We did. We cited previous studies (Dai and Trenberth, 2002; Clark et al., 2015; Ghiggi et al., 2019; Müller et al., 2021) that highlight the Nile as one of the most challenging large-scale basins for hydrological modeling, to underscore the importance of our work in accurately reproducing Nile streamflows (see lines 75–85 in the revised text).

Reviewer 3 Comment 4 (Fig. S1): The colorbar ([0.18–2780] mm/year) does not match the caption (0.018–2780 mm/year). Also, 0.18 mm/year seems unrealistically low. Please double-check these values. We did. We corrected the mismatch between the colorbar and the caption (Refer to the revised Fig. S1 and its caption) by replacing 0.018 with 0.18. The lower bound of 0.18 mm/year is a realistic estimate for the hyper-arid regions in the downstream sections of the Nile Basin, where precipitation is minimal and years may pass without any rainfall.

Reviewer 3 Comment 5 (L125): Please check the text “QQM effectively identified biases in precipitation projections...”. My understanding is that QQM is first applied to the baseline period, not directly to the projections. Do you mean biases in the baseline precipitation, which are then transferred to the projections? We clarified in the revised text that QQM was applied to the 2015–2024 period, when both modeled and observed precipitation data are available, to identify and correct biases in the climate model's precipitation. The derived correction was then applied to the future projections (Refer to lines 124–126).

Reviewer 3 Comment 6 Fig. 3, Fig. 4, and related text: It would be useful also to show precipitation differences between the baseline and projection periods, to better explain the large streamflow increases observed. In particular, I wonder how much precipitation actually increases in the projections to justify streamflow rises of 49% or more. We did. As stated in the original text, our analysis focuses on streamflow extremes—specifically, return-period-based flood frequency analysis—rather than mean flows. In the revised text, we demonstrated that precipitation extremes intensify in the 21st century relative to the historical period (20th century), closely mirroring the substantial increases projected for streamflow extremes. The correspondence between

precipitation and streamflow behavior across the historical and future periods is documented as follows:

- *Historical vs. Future AMS Means.* We compared the mean of the annual maximum series (AMS) generated from daily precipitation for the baseline and future scenarios. The historical mean AMS (1984–2016) extracted from CHIRPS is 9.7 mm/day (Supplementary Fig. 3), whereas the three-model mean increased to 12.5 mm/day under SSP2-4.5 (~29% higher) and 13.7 mm/day under SSP5-8.5 (~41% higher). Refer to lines 136–139.
- *Annual Maximum Series (AMS) of Precipitation.* We calculated precipitation AMS using a 30-day rolling maximum over the entire Nile Basin to ensure comparability with the monthly streamflow AMS (Methods, *Extreme Event Analysis and Uncertainty Analysis*, revised lines 271–273, 410–433, 435–441). The same statistical framework applied to streamflow extremes was extended to precipitation: six candidate probability distributions were tested, fitted using the L-moment method, and ranked using Anderson–Darling tests and L-moment ratio diagnostics. The Log Pearson Type III distribution (LP3) was again identified as the best-fit distribution (Supplementary Table 4; Supplementary Figure 7). The adopted procedures and findings have been incorporated into the revised Supplementary Text 4.
- *Return-Period Analysis of Precipitation Extremes.* Using the fitted LP3 distribution, we generated return-period plots for precipitation extremes under both scenarios, now presented in Supplementary Figure 4 and interpreted in Supplementary Text 3 and in the main text (lines 166–171, 212–213). The comparison shows that precipitation and streamflow extremes intensify in nearly the same proportions. For example, the historical 50-year return-period precipitation was ~186 mm/30 days, rising to ~280 mm (+50%) under SSP2-4.5 and ~338 mm (+81%) under SSP5-8.5. Streamflow shows a highly consistent response, with the 50-year flood (~13,400 m³/s) increasing to ~21,800 m³/s (+63%) under SSP2-4.5 and ~24,700 m³/s (+85%) under SSP5-8.5. Those results confirm that the projected amplification of extreme streamflows is supported by a corresponding increase in precipitation extremes, strengthening confidence in our conclusions.
- *Integration of the above-mentioned findings in the Results and Discussion sections:* We revised the main text to highlight that the projected increase in streamflow extremes in the 21st century is associated with, and corroborated by, a comparable increase in the projected precipitation (Refer to lines 136–139, 166–171, 212–213).

Reviewer 3 Comment 7. Fig. 3: Please specify which catchment the data corresponds to. We did. We revised the Fig. 3 caption to indicate that the AMS projections correspond to the whole study area, aggregated across the four sub-basins.

Reviewer 3 Comment 8. Fig. 4: Please specify which station the data corresponds to; is it Dongola? We did. We state in the caption of Fig. 4 and in the Results section (line 168) that the analysis of extreme streamflow events under the SSP2-4.5 and SSP5-8.5 scenarios was carried out at the Dongola Station. I also suggest showing similar plots for all stations. This would allow readers to infer whether the expected extremes originate from a specific sub-basin. We focus on Dongola because it integrates the upstream contributions of the Blue Nile, White Nile, and Atbara, providing a basin-wide signal most relevant for downstream flood risk. The relative contributions

of these sub-basins have already been evaluated extensively in prior studies, which consistently identify the Blue Nile as the dominant source ($\approx 60\%$; Supplementary text 1), with additional inputs from the White Nile and Atbara (e.g., Beyene et al., 2010; Di Baldassarre et al., 2011; Senay et al., 2014; Badawy et al., 2024). Among the gauges available to us, the Dongola station measures the collective streamflow of the Nile subbasins to downstream regions. Tamaniat mixes White and Blue Nile flows, Khartoum captures a Blue Nile signal, Sennar lies within the Blue Nile, and no long-term measurements are available from the Atbara station. As such, additional return-period plots from these stations would not provide clearer insight into the origin of extremes, whereas Dongola best reflects the integrated downstream hazard emphasized in this study.

Reviewer 3 Comment 9. L163–172: This argument is somewhat weak. The link between the GRACE-observed TWS increase (2003–2023) and the projected 21st-century streamflow rise is not fully convincing. GRACE provides valuable recent observations but does not ensure continuation of the trend. I recommend emphasizing climate model projections of precipitation and runoff as the primary drivers of future streamflow, with GRACE used mainly as supporting evidence for recent consistency rather than as a predictor. We revised the text (lines 174–184) to clarify that GRACE observations are used as supporting evidence of recent consistency rather than as a predictor of future change. We emphasize climate model projections of precipitation and runoff as the primary drivers of the projected 21st-century streamflow increases, while GRACE/GRACE-FO TWS observations (2002–2024) provide independent confirmation that wetter conditions have already emerged in the basin.

Reviewer 3 Comment 10. L209: Which studies? We inserted the intended references directly after the “prior studies” in the revised text (Refer to line 223).

Reviewer 3 Comment 11. L227: Avoid single-sentence paragraphs.

We did. We revised lines 238–240 by merging the sentence beginning with “*Our findings suggest that under SSP5-8.5...*” into the preceding paragraph, ensuring it is no longer a stand-alone sentence.

Reviewer 3 Comment 12. L327–338: Given that NEX-GDDP-CMIP6 already provides bias-corrected variables, could you clarify why an additional bias correction was applied to precipitation? Since CHIRPS is satellite-based and there is sparse in-situ coverage in much of the Nile basin, it would be helpful to discuss potential limitations of this “reference” dataset. In the revised manuscript we added a justification for applying an additional bias correction step beyond the NEX-GDDP adjustments (lines 368–373). Although NEX-GDDP provides bias-corrected variables, residual discrepancies remain in the upper tails of the precipitation distribution. We therefore applied quantile–quantile mapping (QQM) using CHIRPS as the observational reference, ensuring consistency with the dataset used for SWAT+ calibration and improving alignment between the observed and model distributions. Since CHIRPS is satellite-based and there is sparse in-situ coverage in much of the Nile basin, it would be helpful to discuss potential limitations of this “reference” dataset. We expanded our description of CHIRPS (lines 356–365), highlighting its strengths as a high-resolution, widely validated precipitation product that has been extensively applied in Africa, while also acknowledging limitations related to sparse gauge density and complex topography. These points are supported with references to recent validation and review studies (Funk et al., 2015; Dinku et al., 2018; Mulungu & Mukama, 2023; Du et al., 2024). Finally,

to illustrate the effect of the correction, we refer to the revised Results section (lines 128–131) and Supplementary Figure 2, where the CDFs of observed and corrected model data are shown. The figure demonstrates that QQM substantially improved agreement, particularly at the 90th and 99th percentiles, thereby providing a more robust basis for projecting extremes.

Reviewer 3 Comment 13. L334: Please clarify the criteria used to select the 30 GCMs from NEX-GDDP? Also, provide a list of the selected GCMs as supplementary information. As explained in the revised text (lines 350–353), we selected 30 GCMs from the NEX-GDDP dataset out of the 35 models because they include all required variables (precipitation, maximum and minimum temperature, relative humidity, solar radiation, and wind speed) over the Nile Basin for both SSP2-4.5 and SSP5-8.5. As requested, we provided the complete list of the selected 30 models in Supplementary Table 5.

References

- Badawy, A. et al. Floods of Egypt's Nile in the 21st century. *Sci. Rep.* 14, 27031 (2024).
- Beyene, T., Lettenmaier, D. P. & Kabat, P. Hydrologic impacts of climate change on the Nile River Basin: implications of the 2007 IPCC scenarios. *Clim. Change* 100, 433–461 (2010).
- Clark, E. A., Sheffield, J., van Vliet, M. T. H., Nijssen, B. & Lettenmaier, D. P. Continental runoff into the oceans (1950–2008). *J. Hydrometeorol.* 16, 1502–1520 (2015).
- Dai, A. & Trenberth, K. E. Estimates of freshwater discharge from continents: latitudinal and seasonal variations. *J. Hydrometeorol.* 3, 660–687 (2002).
- Di Baldassarre, G. et al. Future hydrology and climate in the River Nile basin: a review. *Hydrol. Sci. J.* 56, 199–211 (2011).
- Dinku, T. et al. Validation of the CHIRPS satellite rainfall estimates over eastern Africa. *Q. J. R. Meteorol. Soc.* 144, 292–312 (2018).
- Du, H., Tan, M. L., Zhang, F., Chun, K. P., Li, L. & Kabir, M. H. Evaluating the effectiveness of CHIRPS data for hydroclimatic studies. *Theor. Appl. Climatol.* 155, 1519–1539 (2024).
- Elshamy, M. E., Seierstad, I. A. & Sorteberg, A. Impacts of climate change on Blue Nile flows using bias-corrected GCM scenarios. *Hydrol. Earth Syst. Sci.* 13, 551–565 (2009).
- Funk, C. et al. The climate hazards infrared precipitation with stations—a new environmental record for monitoring extremes. *Sci. Data* 2, 150066 (2015).
- Ghiggi, G., Humphrey, V., Seneviratne, S. I. & Gudmundsson, L. GRUN: an observation-based global gridded runoff dataset from 1902 to 2014. *Earth Syst. Sci. Data* 11, 1655–1674 (2019).
- Mengistu, D., Bewket, W., Dosio, A. & Panitz, H.-J. Climate change impacts on water resources in the Upper Blue Nile (Abay) River Basin, Ethiopia. *J. Hydrol.* 592, 125614 (2021).
- Müller, O. V., Vidale, P. L., Vannièrè, B., Schiemann, R. & McGuire, P. C. Does the HadGEM3-GC3.1 GCM overestimate land precipitation at high resolution? A constraint based on observed river discharge. *J. Hydrometeorol.* 22, 2131–2151 (2021).

- Mulungu, D. M. M. & Mukama, E. Evaluation and modelling of accuracy of satellite-based CHIRPS rainfall data in Ruvu subbasin, Tanzania. *Model. Earth Syst. Environ.* 9, 1287–1300 (2023).
- Nkwasa, A., Chawanda, C. J., Schlemm, A., Ekolu, J., Frieler, K. & Van Griensven, A. Historical climate impact attribution of changes in river flow and sediment loads at selected gauging stations in the Nile basin. *Clim. Change* 177, 42 (2024).
- Ombadi, M., Nguyen, P., Sorooshian, S. & Hsu, K. Retrospective analysis and Bayesian model averaging of CMIP6 precipitation in the Nile River Basin. *J. Hydrometeorol.* 22, 217–229 (2021).
- Roth, V., Lemann, T., Zeleke, G., Subhatu, A. T., Nigussie, T. K. & Hurni, H. Effects of climate change on water resources in the Upper Blue Nile Basin of Ethiopia. *Heliyon* 4, e00771 (2018).
- Senay, G. B., Velpuri, N. M., Bohms, S., Demissie, Y. & Gebremichael, M. Understanding the hydrologic sources and sinks in the Nile Basin using multisource climate and remote sensing data sets. *Water Resour. Res.* 50, 8625–8650 (2014).
- Takele, G. S., Gebrie, G. S., Gebremariam, A. G. & Engida, A. N. Future climate change and impacts on water resources in the Upper Blue Nile basin. *J. Water Clim. Change* 13, 908–925 (2022).

Review

The study presents a comprehensive, basin-wide assessment of the impacts of 21st-century climate variability on extreme streamflow in the Nile Basin. Using the SWAT+ model forced by bias-corrected CMIP6 GCM outputs under SSP2-4.5 and SSP5-8.5 scenarios, the authors: Predict 63–85% increases in 100-year peak discharges, show that extreme flood events will become more frequent, possibly occurring every decade and emphasize the need for international cooperation in transboundary water governance and promote the use of floods for groundwater recharge opportunities. This paper provides information of great importance to policy makers and water managers in the Nile Basin. I recommend that this manuscript be accepted for publication after some minor comments are addressed.

In some locations, the subfigure labels (e.g., a, b) are bolded, while in others they are not. Please unify the formatting of all figure notations for consistency throughout the manuscript.

The following references appear incomplete and may need full publication details (publisher, editor, ISBN, etc.):

Said, R. *The River Nile: Geology, Hydrology and Utilization* (Elsevier, 2013).

Elsaeed, G. Effects of climate change on Egypt's water supply. In *National Security and Human Health Implications of Climate Change* 337–347 (2012).

Sentence Clarification (Lines 173–175): Our findings align with previous studies that suggested that 20th-century 100-year floods may recur every 2–25 years in a warming climate over the Nile Basin and reported that Africa could see a 152% increase in 50–100-year floods and a 182% increase in 100–200-year floods by the century's end.

Suggested revision: Our findings align with previous studies that indicate 100-year floods from the 20th century could recur every 2 to 25 years under a warming climate in the Nile Basin. These studies also project a 152% increase in 50–100-year floods and a 182% increase in 100–200-year floods across Africa by the end of the 21st century.

Simplify and prioritize the most policy-relevant takeaways in the abstract. End with a strong concluding sentence that highlights the novelty (e.g., “This is the first basin-wide projection study of extreme flows using CMIP6 for the Nile Basin”).

Some sections in methods are too technical, Please consider simplifying the language where possible, particularly in sections related to model calibration, bias correction. Define all abbreviations at their first use.

What does the 5th–95th percentile range mean for policy or flood infrastructure planning?